# The Discovery and Characterization of Conserved and Novel miRNAs in the Different Developmental Stages and Organs of Pikeperch (*Sander lucioperca*)

**DOI:** 10.3390/ijms25010189

**Published:** 2023-12-22

**Authors:** Marieke Verleih, Tina Visnovska, Julien A. Nguinkal, Alexander Rebl, Tom Goldammer, Rune Andreassen

**Affiliations:** 1Institute of Genome Biology, Research Institute for Farm Animal Biology (FBN), 18196 Dummerstorf, Germany; verleih@fbn-dummerstorf.de (M.V.); rebl@fbn-dummerstorf.de (A.R.); 2Bioinformatics Core Facility, Oslo University Hospital, 0424 Oslo, Norway; 3Department of Infectious Disease Epidemiology, Bernhard Nocht Institute for Tropical Medicine, 20359 Hamburg, Germany; julien.nguinkal@bnitm.de; 4Faculty of Agriculture and Environmental Sciences, University of Rostock, 18059 Rostock, Germany; 5Department of Life Sciences and Health, OsloMet—Oslo Metropolitan University, 0167 Oslo, Norway; rune.andreassen@oslomet.no

**Keywords:** micro RNAs, pikeperch, aquaculture, development, qPCR, small RNA sequencing, novel miRNAs, expression pattern

## Abstract

Micro RNAs (miRNAs) are short non-coding RNAs that act as post-transcriptional gene expression regulators. Genes regulated in vertebrates include those affecting growth and development or stress and immune response. Pikeperch (*Sander lucioperca*) is a species that is increasingly being considered for farming in recirculation aquaculture systems. We characterized the pikeperch miRNA repertoire to increase the knowledge of the genomic mechanisms affecting performance and health traits by applying small RNA sequencing to different developmental stages and organs. There were 234 conserved and 8 novel miRNA genes belonging to 104 families. A total of 375 unique mature miRNAs were processed from these genes. Many mature miRNAs showed high relative abundances or were significantly more expressed at early developmental stages, like the miR-10 and miR-430 family, let-7, the miRNA clusters 106-25-93, and 17-19-92. Several miRNAs associated with immune responses (e.g., slu-mir-731-5p, slu-mir-2188-5p, and slu-mir-8159-5p) were enriched in the spleen. The mature miRNAs slu-mir-203a-3p and slu-mir-205-5p were enriched in gills. These miRNAs are similarly abundant in many vertebrates, indicating that they have shared regulatory functions. There was also a significantly increased expression of the disease-associated miR-462/miR-731 cluster in response to hypoxia stress. This first pikeperch miRNAome reference resource paves the way for future functional studies to identify miRNA-associated variations that can be utilized in marker-assisted breeding programs.

## 1. Introduction

Micro RNAs (miRNAs) are short non-coding RNA molecules with an average length of 22 nucleotides (nt). Mature miRNAs are processed from hairpin precursor miRNAs (pre-miRNAs), which, in turn, are Drosha-processed products of transcribed primary miRNAs (pri-miRNAs) [1,2]. MicroRNAs play an important role in the post-transcriptional regulation of their target transcripts. By binding to the target mRNA, the mature miRNA mediates their translational repression or degradation [3]. From studies in mammals, it is estimated that more than 60% of the mRNAs have conserved miRNA binding sites [4]. MicroRNAs are thus involved in the regulation of multiple developmental and physiological processes like growth, immune response, and stress response [4,5,6,7]. In teleost fish, there are similarly high numbers of predicted target genes [8], and the high conservation of mature miRNA sequences suggests shared functions across species [9]. Data on the miRNA repertoire in teleosts are still limited, with only 16 out of the over 26,000 teleost fish species being present in miRBase (22 October 2023; http://www.mirbase.org/). Nevertheless, research on teleostean miRNAs has increased significantly (reviewed in [9,10,11]), as has research on fish models including zebrafish (*Danio rerio*) [12,13] and economically relevant species like Atlantic salmon (*Salmo salar*) [14,15], rainbow trout (*Oncorhynchus mykiss*) [16,17], Nile tilapia (*Oreochromis niloticus*) [18,19], Atlantic cod (*Gadus morhua*) [20,21], lumpfish (*Cyclopterus lumpus*) [22], and channel catfish (*Ictalurus punctatus*) [23,24]. In addition, the recently developed FishmiRNA database also provides miRNAs for black bullhead (*Ameiurus melas*), Japanese medaka (*Oryzias latipes*), molly (*Poecilia mexicana*), European perch (*Perca flavescens*), stickleback *(Gasterosteus aculeatus*), and blackfin icefish (*Chaenocephalus aceratus*) [25]. The studies in fish have, like in higher vertebrates, shown that particular miRNAs are associated with immune response, stress response, and growth and are even supposed to regulate the aforementioned processes. Smaller groups of miRNAs respond, for example, to viral and bacterial infection in Atlantic salmon, rainbow trout, Atlantic cod, and Nile tilapia [20,24,26,27,28]. Many of these miRNAs, including miRNA-146, miRNA-155, and miRNA-223, respond in the same manner to infectious diseases across species, and the predicted targets are key immune genes [10,24,26,27]. Similarly, investigations of the miRNAs associated with smoltification and sea water adaptation in salmonids have revealed groups of miRNAs targeting genes enriched in particular stress and immune response pathways [15,18]. Other studies on teleost fishes like tilapia and zebrafish have identified miRNAs associated with osmoregulatory stress and hypoxia [29,30]. Economically interesting traits like growth are likely regulated in fish by particular miRNAs, as indicated in studies on Nile tilapia [18,19]. Certain miRNAs also respond differently to functional feed designed to elevate robustness to infections. Others are associated with familiar differences in lipid metabolism affecting the fatty acid composition of the fish [31]. Such miRNAs influencing relevant traits may be utilized as biomarkers. Alternatively, differences or variations in expression that affect the miRNA-target gene interaction may be utilized for marker-selected breeding to assure that genetically beneficial variation is included in the breeding stock.

Global aquaculture is experiencing an annual growth of >5%; around 57.5 million tons of finfish are currently produced worldwide, while aquaculture finfish production in Europe is stagnating [32,33]. In order to revive the aquaculture market, production is set to shift towards focusing on high-value fish like pikeperch (*Sander lucioperca*). This predatory freshwater fish belongs to the family of *Percidae* and is found in habitats in western Eurasia. Due to its excellent flesh quality, displayed by a low fat content and delicate flavor without small intramuscular bones, pikeperch has a high consumer acceptance [34,35]. Together with its relatively rapid weight gain, these quality features make pikeperch an economically interesting species for intensive aquaculture systems. The global aquaculture production of pikeperch reached 3074 t of live weight in 2020 (FIGIS FAO 2022, https://www.fao.org/fishery/en/figis (accessed on 13 December 2023)), with a European market value up to USD 22.2/kg (whole fish) in countries such as Germany and France [36]. The production of this percid was established in recirculating aquaculture systems (RAS) in Europe at the beginning of the twenty-first century [37]. Its domestication through successful breeding in aquaculture has progressed steadily in recent years, but several bottlenecks still fundamentally limit the production efficiency [38]. The main problems are the species’ fluctuating reproductive performance and relatively slow specific growth, the low survival rates of the larvae due to cannibalism, high deformation rates or the incorrect development of the swim bladder, feed conversion during early development, and the relatively high associated production costs [38,39]. Although RAS in particular, with their defined environment, are predestined for breeding approaches [40,41], to date, no marker-assisted selection or phenotype-based family selection approaches have been applied in pikeperch breeding programs, which could help to overcome some of the problems. However, in the EU in particular, there has been an increasing shift in recent years towards family-specific and genetic breeding approaches for economically relevant fish species such as Atlantic salmon [42,43]. The provision of complete genetic tools, including genome sequences, SNP sets, or ncRNAs, is essential for future breeding approaches [44]. The exploration of the genetic resources of the species is the backbone for potential marker-based selection. In addition to its growing commercial importance, research on the genetic and genomic composition of pikeperch has also gained momentum. With recent publications on its genome [45,46] and transcriptome [47], the systematic exploration of functional genomics is feasible in the context of, for instance, relevant commercial traits. So far, nobody has explored miRNA genes and their expression in pikeperch. The identification and characterization of the miRNAome in pikeperch is the next step and is crucial to understanding the regulatory gene mechanisms and subsequently applying this knowledge to improve the performance of economically important traits in this fish species. The characterization of pikeperch-specific miRNAs would allow for differential expression analyses involving applying the pikeperch miRNAome in comparisons of the conditions important to fish health or other economically interesting traits. Such miRNAs and any variation affecting their miRNA-target gene interactions have, as mentioned above, the potential to be used as biomarkers and in marker-assisted breeding to benefit pikeperch aquaculture production. Such beneficial variants may also, in the future, be efficiently introduced in breeding stock by gene editing methods.

The aim of this study was to identify the pikeperch miRNA repertoire by using small RNA high-throughput sequencing (HTS). Samples representing different developmental stages and organs were investigated to discover both evolutionally conserved teleost miRNAs and any novel pikeperch miRNAs. Applying comparative approaches, we also aimed to disclose miRNAs likely involved in the regulation of development or those serving organ-specific functions.

## 2. Results and Discussion

### 2.1. Identification of Conserved and Novel miRNA Genes in Pikeperch

Small RNA libraries from 20 pikeperch samples were sequenced. These comprise six different developmental stages and four organs of adult fish. A total of 172.6 million raw reads were obtained. After processing, 94.1 million high-quality and adapter-trimmed reads were left, ranging from 2.5 to 9.2 million reads per sample (Appendix A).

An initial miRDeep2 analysis and subsequent manual characterization led to the identification of 242 miRNA genes in the samples investigated. Most genes were located at single unique locations in the pikeperch genome, while seven were identical duplicated genes positioned in two or three different loci. Basic local alignment search tool (BLAST) homology searches against the stem-loop sequences in the miRNA database could classify 234 of the miRNA genes (96.7%) as orthologues to evolutionary conserved miRNAs. The miRNA genes identified from pikeperch were assigned to 104 families which exhibited a wide range of family members. The let-7 (12 family members) and miR-430 (10 family members) families host the most members in the pikeperch miRNAome. Eight of the miRNA genes showed no match with any of the miRNA genes in miRBase but exhibited all the properties expected for a miRNA gene from our identification criteria (see Section 3.3). Additional nucleotide BLAST (BLASTN) searches against other RNA databases ruled out the notions that they were other kinds of non-coding small RNAs or derived from repetitive elements. Thus, they are the only novel miRNAs observed in pikeperch so far, and they were annotated accordingly. An overview of all miRNA precursor sequences, their genomic location, the sequences of the mature 5p and 3p miRNAs originating from each precursor, and the 5p or 3p abundance differences (arm dominance) is given in Appendix A. The resulting pikeperch miRNAome consists of 375 unique mature miRNA (miRs) sequences (185 mature 5p and 190 mature 3p) (Appendix A). Many of the mature miRNAs were identical, even if processed from different miRNA genes (precursors). The individual annotation in the miRNAome reflects the occurrence of identical mature miRNAs from different loci by adding additional numbers or letters in the suffix of some mature miRNAs. There are, for example, two slu-mir-21 genes (Appendix A), but the mature 5p is identical in both and is thus annotated as slu-mir-21-1-2-5p in Appendix A. Likewise, the two slu-mir-7 genes have mature 5p’s that can not be distinguished, and the resulting mature miRNA was consequently termed slu-mir-7ab-5p in the unique miRNAome.

Pikeperch’s miRNA gene diversity, with 242 identified miRNA genes from 104 conserved families, is similar to the miRNA gene diversity described for other teleosts such as channel catfish (237 miRNAs and 105 families; [23]), lumpfish (391 miRNAs and 104 families; [22]), and Atlantic cod (219 miRNAs and 110 families; [21]). This finding in pikeperch is therefore in accordance with prior studies pointing out that most miRNAs are evolutionarily highly conserved cross-species non-coding RNAs [48,49]. In addition, we also identified eight novel miRNAs in pikeperch. In many studies, including ours, previously unknown or species-specific miRNAs have been identified in teleosts in addition to the orthologous miRNAs. The emergence of novel miRNAs is an evolutionary ongoing process involving several mechanisms [50]. Lu et al. (2008), in their study on the fruit fly (*Drosophila melanogaster*), calculated a high “birth and death rate” for new miRNA genes during evolution [51]. Six of the pikeperch-specific miRNAs (from slu-mir-nov1 and -nov3 to -6 and -nov7) were predominantly expressed in the early fingerling stage; one was also expressed in eyed eggs (slu-mir-nov7), and two novel miRNAs (slu-mir-nov2a and -b) were exclusively abundant in eyed eggs and the early larvae stages (Appendix A). These expression patterns could indicate that their functions are associated with stage-specific development.

Animal miRNAs may be transcribed as individual miRNA genes or, if located in gene clusters, they may be transcribed coordinately [52,53]. These clusters, as defined by miRbase, are composed of at least two physically adjacent miRNA genes (see Materials and Methods) [54]. We identified 113 miRNA genes (47% of all pikeperch miRNA genes) that were grouped into 45 such distinct clusters. Each cluster consists of up to 13 miRNA genes (Appendix A). The gene clusters are localized throughout the pikeperch genome and can be divided into 31 homo- and 14 hetero-clusters. All the miRNA gene clusters discovered in pikeperch, as well as the miRNA genes in the clusters, are evolutionarily conserved and have orthologues in other teleost species (according to miRbase and MetaMirClust, https://fgfr.ibms.sinica.edu.tw/MetaMirClust (accessed on 28 September 2023)). Like in other vertebrates [13,14,21,55], the miRNA-430 gene cluster stands out as the one with the largest number of clustered genes.

### 2.2. Mature miRNA Diversity across Sample Groups and Their Relative Abundance within Sample Groups

Assessing diversity in terms of the different unique mature miRNAs in each sample group revealed the presence of 371 to 375 (all) mature miRNAs in eyed eggs, as well as larvae 2 days post hatching (dph), 15 dph, and 26 dph. The spleen from 3 g- and 35 g-fingerlings contains 363 and 350 miRAs, respectively. The adult organs varied from 325 to 368 different mature miRNAs (Table 1). However, all miRNA genes were expressed at least at a basic level in all samples, as there were at least one mature miRNA (either 5p or 3p) detected from each of the 242 miRNA genes. The exception from this were slu-miRNA-1788 and the two novel miRNAs—slu-miRNA-nov2a and b—that were not detected in spleen fingerling 35 g and liver. Liver also lacked the expression of slu-miRNA-726 and the novel miRNA gene slu-miRNA-nov5. These three novel miRNAs were not detected in gills that also lacked slu-miRNA-430a. A complete overview of normalized read counts for all unique mature miRNAs in all samples is given in Appendix A.

The detection of which mature miRNAs are highly abundant within each sample could provide more information about which miRNAs are biologically important in each group. Such a relative ranking of mature miRNAs by their normalized reads within samples are not hampered by any technical bias as the read numbers used to show the relative abundance are retrieved from the same sample. The read numbers are easily available and can thus directly reflect what the relative abundances of the mature miRNAs in a given sample are. The abundance patterns differed somewhat between (i) the early development from eyed eggs to larvae 26 dph (Figure 1), (ii) the spleen samples from the adult and the two fingerling stages (Figure 2), and (iii) the other organ samples of the adult fish (Figure 3). The top enriched mature miRNAs could be divided into miRNAs that were common to all groups and miRNAs that were highly enriched in certain developmental stages or organs of pikeperch. Figure 1, Figure 2 and Figure 3 display the top 20 most abundant mature miRNAs per sample type (Appendix A). Forty-nine different mature miRNAs from thirty-seven different families dominated the different samples, representing between 62.3% and 82.8% of the total amount of mature miRNAs per developmental stage or organ. Some mature miRNAs, like slu-mir-21-1-2-5p, slu-mir-22ab-3p, and slu-mir-26a-1-3-5p, were ubiquitously enriched throughout the stages. These mature miRNAs also had similar high abundances in all organs and developmental stages in other teleost fish [14,22]. Together, this indicates housekeeping functions that are conserved across species.

While the composition of the top 20 most abundant mature miRNAs in (i) was quite similar between all four developmental stages (Figure 1), some of these mature miRNAs displayed a relatively high abundance in only these four early developmental stage groups (eyed egg to 26 dph), not the other groups ((ii) and (iii)). These were two slu-mir-10 family members (slu-mir-10b and -c-5p), slu-mir-9-1-4-5p, slu-mir-7ab-5p, slu-mir-1-3p, and slu-mir-206-3p. This could reflect their importance in the regulation of early development. The findings from an analysis of Atlantic salmon support the notion that the mature slu-mir-10b and -c-5p have such important developmental functions, as they constituted more than 50% of the total number of miRNAs in the early developmental stages of Atlantic salmon [14]. In line with this is the finding that they regulate hox genes in Nile tilapia embryonic development [56] and also mediate cell proliferation and differentiation [57]. The mature members slu-mir-9-1-4-5p and slu-mir-7ab-5p have been reported as highly abundant in early developmental stages, as well as in the brain [14,22,58,59,60,61]. It is likely that their high relative abundance in early developmental groups in pikeperch reflect similar functions. Studies have also revealed that orthologs of slu-mir-1-3p and slu-mir-206-3p are important in muscle development and embryonic development [62,63], which would also agree with their high relative abundance observed in the pikeperch developmental groups.

Slu-mir-143-3p was by far the most abundant mature miRNA in the two spleen samples from the fingerling stages, making up 1/6 of the total miRNA read counts in juvenile pikeperch (~17 and ~18%, Figure 2). This indicates that it has a relative abundance in the fingerling spleen that is more than three times higher than what is seen in any other organ. A similarly high abundance was detected in adult spleen (11%). In addition, slu-mir-144-5p displayed a high relative abundance in organ samples from the spleen, while slu-mir-8159-5p, slu-mir-101a-3p, and both of the mature miRNAs processed from slu-miRNA-2188 were predominantly enriched in the spleen of the juvenile fish. The phase of larvae and fingerling stages in fish is correlated with high growth rates and elevated energy demands. Studies on the regulatory role of the miR-143 family in human and fish link it to lipid metabolism [64,65]. In rainbow trout, an increase in miR-143 abundance during early development up to the juvenile stage has been demonstrated [65]. Mir-144-5p shows similarly high abundance in Atlantic cod spleen [21] and is associated with erythropoiesis [66]. Together with miR-101-3p, it is associated with macrophage function [67]. miRNA-2188, which is teleost-specific, is associated with immune responses and monocyte to macrophage maturation in fish [10,27], while the other teleost-specific miRNA that was highly abundant in the fingerling spleen, miR-8159-5p, is associated with immune response in fish [26,68].

The relative abundance of some mature miRNAs in adult pikeperch seems to differ between organs (Figure 3). Remarkably, slu-mir-122-5p accounted for more than one fourth of the total miRNA read count in the liver. Similarly high abundance has been seen in liver samples in all vertebrates, including teleost fish [14,21,22,69]. MicroRNA-122-5p was demonstrated to be a key regulator of cholesterol metabolism in the liver [69]. In addition, slu-mir-192-5p was among the top 20 abundant mature miRNAs in the liver and in the larva developmental stages. Such a high liver abundance has also been reported in other teleost fish [14,21]. In agreement with the notion of it having both developmental and liver-specific functions, it has been reported that miR-192-5p is abundant in the liver of mammals, where it promotes the development of the liver and coordinates energy metabolism [21,70].

Slu-mir-205-5p is highly abundant only in gills. This gill-specific enrichment is also conserved in other fish species [22,29,71]. However, its function in gills is not well known. Similarly, slu-mir-203a-3p is highly abundant in gills and in the developmental stages. Again, this has been reported in other fish species [14,22], but its function is not known. The observed high variation in the relative abundance of mature miRNAs between the different developmental stages and organs suggest that miRNAs are important in organ and tissue differentiation and/or the maintenance of their functions. The conserved miRNAs showed similar abundance patterns in other species, as observed in pikeperch, strongly suggesting that they serve the same conserved functions in pikeperch as well.

### 2.3. Differential Expression Analysis and qPCR Revealed Additional miRNAs Associated with Particular Life Stages and Organs

The biologically important mature miRNAs at particular life stages or in the different organs are not necessarily the highly abundant ones. Thus, to further investigate whether some mature miRNAs are differentially expressed during pikeperch development beyond what was obvious from the distribution of the most abundant ones, we performed DESeq2 expression analyses including comparisons between (i) the early developmental stages, (ii) the two fingerling stages, and (iii) the adult organs.

The selection of the differentially expressed mature miRNAs detected by the DESeq2 analyses was validated by qPCR, along with some mature miRNAs identified as particularly abundant in a given sample (see Section 2.2) (Appendix A). To identify stable reference miRNAs, the expression levels of four candidate reference miRNAs were analyzed by qPCR in all 48 samples, and RefFinder suggested that slu-mir-30c-5p is the most stable single gene (stability value of 1.32) and that slu-mir-30c-5p/slu-mir-30-1-2-5p was the best two-gene combination (stability value of 0.58). The two reference miRNAs mir-107-3p and mir-26a-1-3-5p, with proven high stability in Atlantic salmon, were rated as less stable across the samples tested in pikeperch [14,72]. The two genes in the two-gene combination with superior stability were therefore subsequently used as reference genes for our qPCR analysis. The sample-specific high abundances and significant differences from the DESeq2 analysis were mostly confirmed in the qPCR analysis (Figure 2, Appendix A). Furthermore, there was a strong positive correlation between the results from DESeq2 and qPCR (Pearson’s correlation coefficient, r = 0.9) (Figure 4a–c). This strong correlation indicates that the other DESeq2 results, although not validated by qPCR, are also reliable.

A differential expression analysis, facilitated by DESeq2 and involving the comparison of the very early stages and larvae (eyed egg and 2 dph vs. 15 dph and 26 dph), led to the identification of 98 differentially expressed mature miRNAs (Table 2, Appendix A). This large number was not unexpected, considering the major anatomical changes, the specialization of tissue, and the development of organs occurring in this early phase [28,73,74]. The very early stages revealed 44 mature miRNAs from 25 families, with significantly higher expression compared to the larvae stage. Among these were slu-mir-9-1-4-5p and the miRNA 10 family of mature 5p miRNAs. These displayed a generally higher relative abundances in all the developmental stages compared to the adult organ groups (see Section 2.2). Here, the analysis between early developmental stages also revealed that their expression levels were highest in the very early stages. Another family standing out as highly expressed was the miRNA 430 family (fold change > 50×). This was not unexpected, as they are known to promote the removal of maternal mRNAs [55], and similar expression patterns were also observed in Atlantic salmon in [14]. The clustered miRNAs 25, 93, and 106 also showed an increased expression of their mature miRNAs at this very early stage. This miRNA cluster is known to control the epithelial to mesenchymal transition in vertebrates [75,76]. Another miRNA gene cluster, 17-19-92, was significantly enriched at this very early stage. Similarly high expression was reported in vertebrate embryos in [77], wherein they were associated with skeletal development. The deletion of the two clusters 106-25-93 and 17-19-92 is lethal to embryos in mammals [77,78].

On the other hand, the let-7-5p mature miRNAs was strongly increased in expression (fold changes from 3–20×) in the larvae stage compared to the very early stages. This is in agreement with their function as regulators of the entry into the larvae stage [74,79,80]. We also noted a higher expression in the larvae of slu-mir-133ab-3p, which are clustered with miR-1 and miR-206 (Appendix A) and act together with these miRNAs to control muscle development (see Section 2.2) [81]. Both the 5p and 3p mature miRNAs of the slu-mir-199 genes were highly increased in the larvae stages, in accordance with findings for other teleost fishes ([14,21,22,82,83]). This miRNA controls sonic hedgehog signaling, which is essential to normal embryonic development [84]. It is clustered with the miRNA-214 genes that are also enriched in larvae. These two miRNAs have also been reported as important in skeletal development in zebrafish [85]. Four other mature miRNAs—the clustered mature miRNAs slu-mir-143-3p and slu-mir-145-5p and the clustered matures slu-mir-192-5p and slu-mir-194b-5p—were also particularly increased in the larvae stage. Again, this seems to be conserved in fish [14,22], and their function in vertebrate development is associated with cardiac progenitor cell differentiation and muscle development, respectively [86,87]. One teleost-specific miRNA, slu-mir-8159-5p, was increased. Despite being associated with immune system and stress response [26], the particular function of slu-mir-8159-5p, at this stage, is not known.

The early development of teleost fish, from fertilized egg, through the larval stages, to juvenile fish, involves a series of major anatomical changes, including, for example, the growth of muscles and the skeleton, the appearance of fins, the development of the nervous system, and the adjustment of the digestive system (as reviewed in [73]). MicroRNAs are instrumental in regulating these anatomical changes, exhibiting high levels of tissue- and cell-specific abundance [28]. Many of the conserved pikeperch mature miRNAs, as well as many of the clustered genes, displayed similar enriched expression profiles, as in other vertebrates at embryo/larvae life stages. This finding suggests they also share the same important regulatory functions in early development, as revealed in other vertebrates. The joint increase in the mature pikeperch miRNAs annotated as clustered genes (Appendix A) supports the notion that they are transcribed as polycistrons. The detection of the similar expression changes in such polycistrons also lends credence to the expression analysis itself, as each of the mature miRNAs were independently analyzed in the DESeq2 pipeline.

When comparing miRNA expression in the two fingerling stages from spleen, 13 upregulated miRs from 10 families were found in the early stage of 3 g fingerlings. Some of these were also seen in the relative abundance investigation (Section 2.2). Interestingly, one of these, slu-mir-nov6-3p, was among the novel pikeperch miRNAs characterized in this study. Two other miRNAs, slu-mir-2188-5p and slu-mir-8159-5p, are associated with immune system and pathogen responses in several fish species [20,26,68], while a third highly expressed miRNA, slu-mir-216a-1-2-5p, is a conserved mature miRNA that is reportedly associated with viral infection in rainbow trout spleen [88]. The late stage of 35g fingerlings spleen depicted 6 miRNAs from different families as highly expressed. The significantly increased expression of slu-mir-1388-3p and slu-mir-731-5p was also confirmed by qPCR (Figure 2). Orthologs of all these six mature miRNAs (slu-mirs-1388-3p, -730-5p, -7132-5p, -731-5p, -125-1-3-5p, and -99b-5p) have also been associated with pathogen response in fish [10,26,27,88,89]. In summary, most miRNAs with particularly high expression in one of the two fingerling spleen samples were associated with the function of spleen as an important immune organ in fish [90].

The comparisons between adult organs were not only in agreement with the relative abundance distributions (Section 2.2) but also revealed some additional miRNAs that were preferentially expressed in some organs (Appendix A). There was, for example, an up to 165-fold higher expression of mir-10 family members in the head kidney (slu-mir-10b-5p, -10c-5p) compared to the other organs (Table 2). The enrichment of this family in this central immune organ of bony fish has already been demonstrated in studies on codfish (*Gadidae*) [21], snakehead (*Channidae*) [91], and carps (*Cyprinidae*) [67,92]. The mir-10 family seems to play an important regulatory role in the inflammation and innate immune response of vertebrates and invertebrates through the post-transcriptional regulation of Toll-like receptor signaling and associated transcription factors [93,94].

The abundance of the three most highly expressed mature miRs in the gills (slu-mir-205-5p, slu-mir-203b-3p, and slu-mir-725-3p) was up to 440-fold higher compared to the expression levels in the other organs (Table 2). This organ-specific enrichment is consistent with the findings of many other studies on teleosts and mammals [21,22,29,69,95]. The DE of the dominant 5p-arm of slu-mir-205-5p in the gills, as well as slu-mir-122 in the liver, was also confirmed by qPCR (Figure 3). The liver-enriched miR-122 is, as mentioned in Section 2.2, a key regulator of lipid metabolism in mammals and fish. Menningen et al. (2014) concluded that the involvement of miR-122 in liver-specific metabolic processes could have arisen evolutionarily early in vertebrates [96].

### 2.4. The miR-462/miR-731 Cluster Is Associated with Hypoxia Stress in Pikeperch

The teleost-specific cluster miR-462/miR-731 is involved in pathogen-associated immune responses and the response to hypoxia stress in bony fish [10,30,97,98]. Previously, we studied the influence of chronic oxygen deficiency on the immune status of pikeperch [99]. To further evaluate the possible conservation of the hypoxia-related functions of the miR-462/miR-731 cluster in pikeperch (Appendix A), we investigated the expression of the clustered miRNAs under low-oxygen saturation. Maintaining adult pikeperch for one day at low dissolved oxygen (DO) levels (±3.2 mg/L DO) significantly increased the miR-462/miR-731 cluster expression by 0.4-fold compared to that at normoxic levels (±8.3 mg/L DO) (Figure 5). Huang et al. (2015, 2017) found a similar HIF1α-induced expression of the respective cluster members in zebrafish larvae and adult blunt snout bream (*Megalobrama amblycephala*) 4–24 h after exposure to hypoxic conditions [30,97].

## 3. Materials and Methods

### 3.1. Fish Sample Material

The early-stage pikeperch used in this study were bred and reared in RAS at the State Research Centre for Agriculture and Fisheries Mecklenburg-Vorpommern (Hohen Wangelin, Germany); the fingerlings and adult individuals were bred and reared at the Experimental Animal Facility for Aquaculture of the Institute for Farm Animal Biology (FBN, Dummerstorf, Germany). Pikeperch progeny was reared in a RAS system with a total volume of 9 m^3^. Larvae were kept in fish tanks with a volume of 0.5 m^3^ at an initial stocking density of 100 larvae/L with a water exchange rate of 20%/h/tank, water temperature of ~15.7 °C, and ~9.2 mg/L dissolved oxygen (DO). The water quality was ensured by continuous purification, UV disinfection, and the daily monitoring of relevant water quality parameters. The photoperiod during hatchery was set at 24 h light (L): 0 h dark (D) until day 45 and subsequently at 17 L:7 D (+1.5 h dusk and dawn). The fingerlings and adult fish were kept in 160 L round tanks at 21.1 °C and ~8.1 mg/L DO and subjected to a 12 L:12 D cycle until sampling.

A total of 52 samples were used for miRNA sequencing and quantitative expression analysis using real-time quantitative PCR (qPCR) (Appendix A). Samples were selected to cover as many life stages as possible, taking into account the known critical phases of pikeperch rearing mentioned in the Introduction section. This included eyed eggs (six pools of n = 10 each), yolk sac larvae (two days post hatching (dph) and 15 dph: each six pools of n = 3), and larvae (26 dph: six pools of n = 3 each). Furthermore we sampled spleen from two fingerling stages (3g and 35g: n = 6 each), as well as the liver, gills, head kidney, and spleen of four 19-month old adult individuals each (n = 4 each). The sampling and killing methods followed the standards described in the German Animal Welfare Act [§ 4(3) TierSchG]. In detail, the pikeperch were euthanized with an overdose of 2-phenoxyethanol (50 mg/L water), followed by a blow to the head and spine sectioning at the skull level. The collected material was snap-frozen in liquid nitrogen and stored at −80 °C until further investigation. Of the total 52 samples, 2 samples of the developmental stages and adult organs were used for the high-throughput sequencing of the miRNAs (total of 20 samples). The remaining four samples from each of the different developmental stages were used for qPCR expression analysis (total of 24 samples), together with the four organ samples from each of the adult fish (total of 16 samples, including the two samples of each organ also used for HTS (Appendix A).

In addition, we used eight additional organ samples collected in a prior study by our working group to analyze the expression of the miR-462/731 cluster in response to hypoxia stress [99]. Four pikeperch spleen samples subjected to normal (±8.3 mg/L dissolved oxygen (DO)) or low-oxygen conditions (3.2 mg/L DO) for one day were used for our expression analysis (Appendix A).

The dissection and sampling protocol was approved by the Committee on the Ethics of Animal Experiments of Mecklenburg-Western Pomerania (Landesamt für Gesundheit und Soziales LAGuS; approval ID: 7221.3-1-009/19).

### 3.2. Small RNA Isolation, Library Preparation, and Sequencing (Ilumina Technology)

Total RNA was isolated from the 20 samples by using the mirVana miRNA isolation kit (Thermo Fisher Scientific, Bremen, Germany) following the manufacturer’s protocol. The integrity and quality of the total RNA was evaluated using Nanodrop and an Agilent Bioanalyzer 2100 platform (Agilent Technologies, Waldbronn, Germany), employing the Small RNA assay (Agilent Technologies). Only RNA samples with RNA integrity values (RIN value) ≥ 7 were further processed (Appendix A).

The small RNA libraries were constructed by using the NEBnext^®^ multiplex small RNA Library Prep Set (New England Biolabs, Inc., Ipswich, MA, USA) according to the manufacturer’s protocol. One ug of total RNA was used as the input for the preparation of the libraries. The library preparation included 5′ and 3′ adapter ligation, reverse transcription, and a size selection of 140–150 bp fragments using 6% polyacrylamide gel. An Illumina NextSeq 500 providing 75 bp single end reads was used for sequencing, which was carried out at the Norwegian Sequencing Centre (NSC).

Raw data from the deep sequencing of the samples have been submitted to the National Center for Biotechnology Information (NCBI) Sequence Read Archive (SRA) database under the BioProject accession number PRJNA977610.

### 3.3. Pre-Processing and miRDeep2 Analysis of Sequencing Data

The quality of the sequenced small RNA raw reads was checked using FastQC (v.0.11.8). The adapter sequence (5′ AGATCGGAAGAGCACACGTCTGAACTCCAGTCAC 3′) was removed using Cutadapt (v.2.3) Python Package (v.3.7.3) [100]. This was followed by the size filtering of the raw reads. Reads that were shorter than 18 nts or longer than 25 nts were discarded. An additional FASTQC analysis after size filtering ensured that the per base sequence quality had phred scores of 32 or more in all samples. The processed high-quality reads were used for downstream analysis applying the miRDeep2 software package (v.2.0.0.7) [101]. The present version of the pikeperch genome assembly (GeneBank accession number: GCA-008315115.2) was used as a reference genome for miRNA identification. All samples were processed independently. During processing, one liver sample was removed due to the uncertainty of its origin (possible contamination). The workflow applied to identify conserved and novel pikeperch miRNA sequences as described in Woldemariam et al. (2019) [14]. In short, it comprises the following: The miRDeep2 tool-specific log-odds score (miRDeep2 score) of ≥1.0 × 10^2^ was used as a cut-off for the detection of the miRNA precursor and the two mature miRNAs (5p and 3p) to prevent the inclusion of false positives. All those with scores above the threshold were thus regarded as putative precursors with their mature miRNAs processed as expected from Dicer cleavage. They were additionally inspected to fulfil the following criteria: (1) detected in at least two independent samples, (2) at least 10 sequence reads of 3p and 5p mature miRNAs mapped without mismatches to the hairpin precursor, (3) the mature miRNAs were paired with the precursor hairpin with 0–4 nt overhang at their 3′ ends, (4) the reads mapped supported a consistent pre-processing of 5′ end (5′ homogeneity), and (5) more than 60% of the bases of the mature sequences paired in the hairpin structure [54]. All miRNA precursor sequences were blasted against the miRNA database (miRNBase, version 22.1) (https://www.mirbase.org/search.shtml (accessed on 8 March 2022)). In the case of returning a significant hit (e-value of ≤1 × 10^−5^) against any hairpin precursor in the database, they were annotated as a conserved pikeperch miRNA orthologue and further annotated by us according to the miRbase nomenclature guidelines (slu- prefix and the same number as the orthologues in other teleosts) [54,102]. The precursors that fulfilled our selection criteria but did not significantly match any of the miRNbase entries were considered as putative novel miRNAs. They were further analyzed via blastn searches against known RNA databases in GeneBank (http://blast.ncbi.nlm.nih.gov/Blast (accessed on 15 March 2022)), Rfam v.14.9 (https://rfam.xfam.org/search (accessed on 15 March 2022)), RefSeq (Last update: 4 December 2020 https://www.ncbi.nlm.nih.gov/refseq/), and fRNAdb v.4.0 (https://dbarchive.biosciencedbc.jp/en/frnadb/desc.html (accessed on 15 March 2022)). If any of these putative novel precursors showed significant hits against any other kind of smallRNAs, they were removed from the dataset. Finally, the remaining putative novel miRNAs were used as queries in blastn searches against the pikeperch reference genome. Any putative precursor that provided significant BLAST hits (E-value ≤ 1 × 10^−5^) against multiple loci (>5) in the pikeperch genome were considered repetitive sequence elements and discarded from our dataset. Those that passed all these filtering fit the criteria expected for miRNAs and, consequently, were annotated as novel pikeperch miRNAs.

### 3.4. Identification of miRNA Gene Clusters

We identified miRNA clusters in pikeperch by comparing the precursor locations within contigs according to the following miRBase definition: (1) two or more miRNAs located no more than 10 kilo bases (kb) apart, (2) the miRNAs are transcribed in the same orientation, and (3) are not separated by a transcription unit in the opposite orientation.

### 3.5. Prediction of Enriched and Putative Differently Expressed (DE) miRNAs in and among Developmental Stages and Organs

Comparisons to indicate the mature miRNAs highly expressed in certain organs or between developmental stages were carried out using the miRNAome from this study as described in [22]. In short, processed reads from all 20 samples were mapped to the unique mature miRNAs in the miRNAome characterized in this study by applying STAR aligner software (v.2.5.2b). The index for mapping was thus generated from the unique mature pikeperch miRNAs discovered in this study (Appendix A) with the parameter genomeSAindexNbases 6. STAR aligner software (v.2.5.2b) with alignIntronMax1 and default parameters was then used for the mapping of reads. Next, the STAR mapping BAM output files were processed further in R-Studio by using the featureCounts function from the Rsubread package (v.1.34.2) to produce count matrices. The count tables were used as inputs in the DESeq2 R package (v.1.24.0) for differential expression analysis.

For statistical valid expression analysis, we pooled counts from eyed eggs and larvae 2 dph (n = 4) and larvae from 15 dph and 26 dph, (n = 4), respectively. For a comparison of the two fingerling stages (n = 2 vs. n = 2), we combined the DESeq2 prediction, with verification being carried out via subsequent qPCR analysis. To improve the reliability of the DESeq2 comparisons between the organ samples from adult fish, each organ was analyzed against a pool of the other three (n = 2 vs. n = 6), and some results were additionally verified by qPCR. Liver was not included in the DESeq2 analysis as one of the two samples did not meet our quality measures.

Only miRNAs with normalized read calls of ≥30 in the samples compared were included in the analysis. Differential expression between the samples was tested using Wald’s log2 fold change test. MiRNAs with a log2 fold change threshold of at least ≤−1.0 or ≥1.0 and a Benjamini–Hochberg adjusted *p*-value of ≤0.05 were considered differently expressed. It is well known that, often, only one of the mature miRNAs originating from a precursor is incorporated into Argonaut (RISC complex), while the other is simply degraded. This leads to large disproportions in the read numbers of two mature miRNAs (5p and 3p) originating from the same precursor [103]. The one incorporated into Argonaut is often referred to as the guide-miRNA, while the other is named a passenger-miRNA. As the abundance of such mature pairs from the same precursor was easily accessible in our data, we also filtered out the ones with a clear indication that they were the mature passenger miRNA. This filtering used the following criteria: if a mature miRNA was 10 times less abundant than the dominant, assumed biologically relevant guide miRNA, it was not reported among our differentially expressed mature miRNAs (Appendix A).

### 3.6. Stability Testing of Possible Reference miRNAs and RT-qPCR Analysis

The normalized read counts of the mature pikeperch miRNAs were compared across samples to search for reference genes suitable for use as normalizers in the subsequent quantitative real-time PCR analysis (qPCR). The relative standard deviation (RSD, coefficient of variation) of each mature miRNA was calculated and used as a measurement of their stability across the samples used for deep sequencing. The four predicted reference miRNAs with a moderate or high abundance (between 2500 and 130,000 normalized read counts) and a low RSD (≤0.3)—slu-mir-30c-5p, slu-mir-107-3p, slu-mir-26a-1-3-5p, and slu-mir-30-1-2-5p—were chosen, and 48 samples were analyzed by qPCR; the results were used for subsequent stability testing using the analytic tool RefFinder (http://www.leonxie.com/referencegene.php (accessed on 14 September 2022)) [104]. This software combines the output of the programs gNorm [105], Normfinder [106], BestKeeper [107], and Delta-Ct [108]. RefFinder builds the geometric means of the individual rankings of the four tools and thereby classifies the suitability and reliability of the candidate reference miRNAs. The analysis classified slu-mir-30c-5p as the most stable single gene (stability value of 1.32) and slu-mir-30c-5p/slu-mir-30-1-2-5p as the best two-gene combination (stability value of 0.58). This two-gene combination was used as a reference in the qPCR analysis.

The ten mature miRNAs (slu-mir-205-5p, slu-mir-122-5p, slu-mir-1388-3p, slu-mir-731-5p, slu-mir-1388-5p, slu-mir-551-p, slu-mir-202-5p, slu-mir-10b-5p, slu-mir-130a-5p, and slu-mir-130ab-3p) that were predicted to be significantly more expressed or enriched in one or more of the samples were selected for further expression analysis by real-time qPCR to verify the DESeq2 results. In addition, we analyzed the expression profiles of the mature miRNAs slu-mir-462-5p and slu-mir-731-5p to investigate the expression of the miRNA cluster miR-462/731 during hypoxia stress. A list of the primers used in this study is provided in Appendix A.

Expression analyses were performed on a LightCycler96 system (Roche Diagnostics, Mannheim, Germany) using the SensiFAST^TM^ One-Step qPCR kit (Bioline, Luckenwalde, Germany) in line with the manufacturer’s instructions. Therefore, total RNA (500 ng) was polyadenylated and reverse-transcribed using the miRNA 1st-Strand cDNA Synthesis Kit (Agilent Technologies) according to the manufacturer’s instructions. The resulting cDNA was diluted with 150 µL of RNase-free water and stored at −80 °C until further use. A RT-qPCR reaction was performed using a miRNA-specific forward primer in combination with a universal reverse primer (Agilent Technologies) and 35 cycles of the following reaction conditions: initial denaturation at 95 °C for 5 min, denaturation at 95 °C for 15 s, annealing at 54/60 °C for 10 s, and extension at 72 °C for 20 s. Changes in miRNA expression were calculated based on the ∆Cq quantification model, including the calculation of the primer efficiencies and relative quantity (RQ) normalization using the geometric mean of the two selected reference genes, slu-mir-30c-5p/slu-mir-30-1-2-5p [105,109]. Samples with Cq values higher than 30.0 were excluded from the downstream calculations. For size and quality control, the PCR products were verified via gel electrophoresis. A melting curve analysis validated the amplification of distinct products. Statistical significance (*p* < 0.05) was evaluated by performing a parametric *t*-test.

### 3.7. Statistical Analysis

The qPCR data were analyzed using LightCycler 96 software v. 4.0.1. The graphs were generated using GraphPad Prism (version 9.5.1) software. qPCR data are presented as the means + SEM containing four replicates.

## 4. Conclusions

In this study, we identified the pikeperch miRNA repertoire, which comprises 234 conserved and 8 novel miRNAs. We profiled their abundance during development (from eyed eggs to the organs of adult fishes). While the expression profiles of many miRNAs were similar in all materials, indicating that they regulate housekeeping cellular functions, others showed developmental and organ-specific expression, indicating the regulation of specialized biological functions. The many similarities in expression and the abundance of certain miRNAs at particular developmental stages or organs with other teleosts evidence that these miRNAs maintain important shared evolutionary conserved regulatory functions. The knowledge gained on the miRNAs in this species paves the way for the further exploration of individual miRNAs by qPCR or HTS analysis to disclose their association with traits important to aquaculture production. The identification of miRNAs associated with traits such as development and growth could contribute to tackling the current problems connected to the critical life stages during pikeperch rearing in aquaculture. These specific miRNAs have the potential to be applied as biomarkers in breeding programs involving the application of marker-assisted selection, especially if family-specific expression differences or genetic variation affect the miRNA–target gene interaction. This could further be used for targeted modifications, such as gene editing (e.g., CRISPR), that may effectively modify traits of interest.

## Figures and Tables

**Figure 1 ijms-25-00189-f001:**
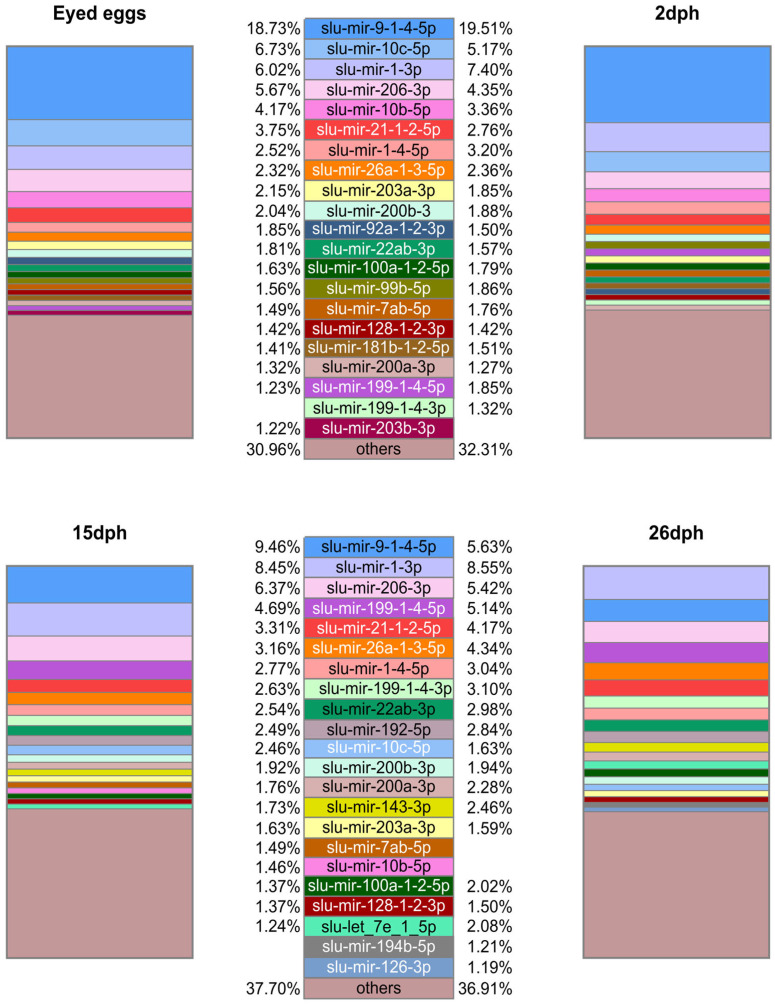
Top 20 most abundant mature miRNAs in pikeperch. The boxes show the scaled proportion (%) of mature miRNAs in two respective replicates of eyed eggs and three larvae stages (two days post hatching [dph], 15 dph, and 26 dph). Each mature miRNA is labeled in a specific color given in the legend.

**Figure 2 ijms-25-00189-f002:**
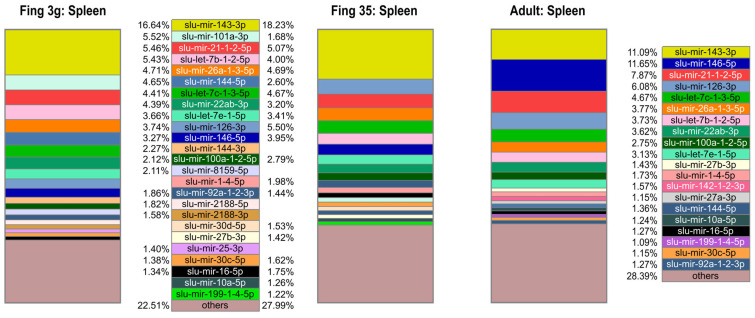
Top 20 most abundant mature miRNAs in pikeperch. The boxes show the scaled proportion (%) of mature miRNAs in two respective replicates of spleen tissues from two fingerling stages (3 g and 35 g) and adult fish. Each mature miRNA is labeled in a specific color given in the legend.

**Figure 3 ijms-25-00189-f003:**
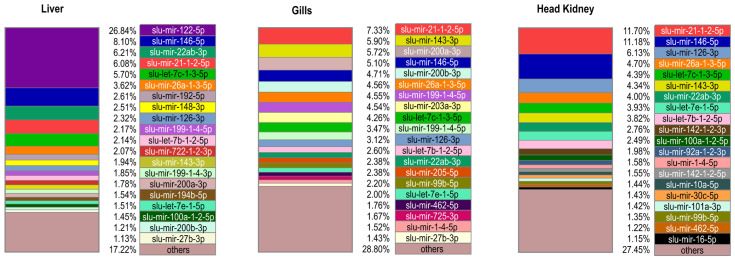
Top 20 most abundant mature miRNAs in pikeperch. The boxes show the scaled proportion (%) of mature miRNAs in two respective replicates of head kidney and gill, as well as of one adult fish liver sample. Each mature miRNA is labeled in a specific color given in the legend.

**Figure 4 ijms-25-00189-f004:**
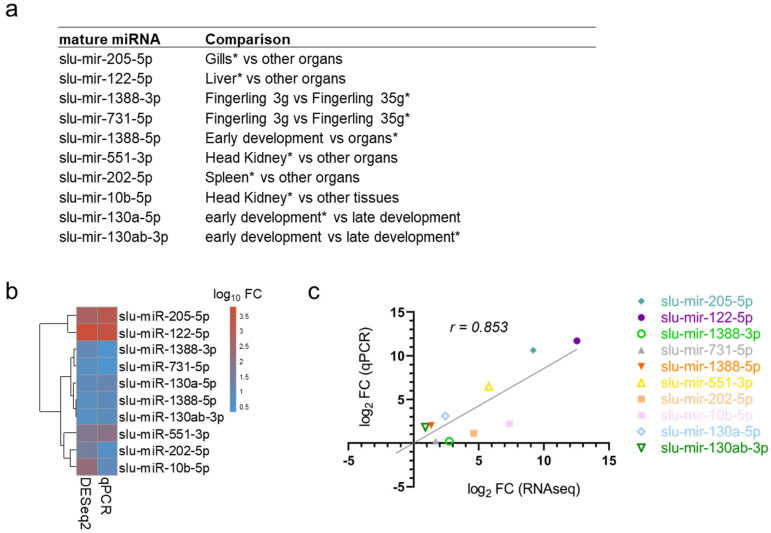
The validation of miRNA expression data by qPCR. (**a**) The table lists ten selected affected mature miRNAs and the compared samples. * developmental stage/organ in which the miRNA is predicted to be significantly enriched (adjusted *p*-value of ≤0.05, Log2FC ≥ 1). (**b**) Heatmap displaying differential expression (log10 fold change) based on norm. reads/DESeq2 and qPCR data. (**c**) Pearson correlation between small RNA-seq (x-axis) and qPCR (y-axis) data for the selected miRNAs. Symbol shape and colors correspond to the respective miRNAs listed beside the plot.

**Figure 5 ijms-25-00189-f005:**
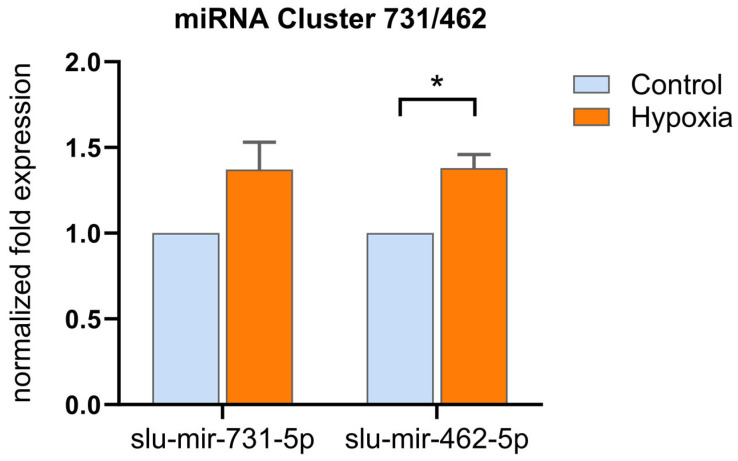
Expression of slu-mir-731-5p and slu-mir-462-5p in response to hypoxic stress (evaluated by qPCR). The columns represent the n-fold expression (means + SEM, n = 4) under controlled (light blue) and hypoxic (orange) conditions, normalized to two reference miRNAs. Significant differences in expression levels between the conditions are indicated by asterisks (Student’s *t*-test; * *p* < 0.05).

**Table 1 ijms-25-00189-t001:** Distribution of unique mature miRNAs in pikeperch by developmental stage and organ.

Sample Group	miRs (Total)
Eyed eggs	374
Larvae 2 dph	375
Larvae 15 dph	372
Larvae 26 dph	371
Spleen Fingerling 3 g	363
Spleen Fingerling 35 g	350
Head Kidney	368
Spleen	366
Liver	325
Gills	357

**Table 2 ijms-25-00189-t002:** Top ten differentially expressed mature (assumed guide) miRNAs in pikeperch based on comparative DESeq2 expression analysis.

miRNA		Fold Change	Adj. *p*-Value
Eyed eggs and larvae 2 dph vs. larvae 15 dph and 26 dph			
*Enriched in eyed eggs and larvae 2 dph:*			
	slu-mir-430b-2-3p	268.78	7.66 × 10^−26^
	slu-mir-430d-1-2-3p	235.77	1.43 × 10^−26^
	slu-mir-430bc-3-3p	222.98	9.14 × 10^−27^
	slu-mir-430bc-2-3p	181.61	4.02 × 10^−24^
	slu-mir-430c-1-3-5p	73.14	4.10 × 10^−18^
	slu-mir-430b-1-4-5p	64.95	9.95 × 10^−15^
	slu-mir-430a-5p	59.58	2.55 × 10^−15^
	slu-mir-nov2ab-5p	57.85	1.68 × 10^−15^
	slu-mir-nov2ab-3p	21.87	1.85 × 10^−15^
	slu-mir-219a-2-3p	20.55	5.57 × 10^−18^
*Enriched in larvae 15 and 26 dph:*			
	slu-let-7d-1-5p	20.04	9.67 × 10^−17^
	slu-let-7f-1-5p	17.99	1.02 × 10^−14^
	slu-let-7h-1-5p	17.38	4.83 × 10^−14^
	slu-mir-726-3p	15.09	4.52 × 10^−23^
	slu-mir-194a-5p	12.49	1.59 × 10^−23^
	slu-mir-31-5p	12.05	1.40 × 10^−16^
	slu-mir-726-5p	11.41	1.37 × 10^−14^
	slu-mir-146-5p	9.40	1.21 × 10^−8^
	slu-mir-194b-5p	9.33	1.67 × 10^−13^
	slu-let-7b-1-2-5p	8.59	3.52 × 10^−10^
Spleen fingerlings 3 g versus Spleen fingerlings 35 g			
*Enriched in fingerling 3 g*			
	slu-mir-nov6-3p	50.40	1.60 × 10^−3^
	slu-mir-183-1-2-5p	46.21	2.02 × 10^−5^
	slu-mir-8159-5p	20.85	3.15 × 10^−3^
	slu-mir-216c-5p	19.59	1.91 × 10^−2^
	slu-mir-216b-5p	19.50	1.91 × 10^−2^
	slu-mir-216a-1-2-5p	15.50	2.50 × 10^−2^
	slu-mir-375-3p	15.35	8.39 × 10^−4^
	slu-mir-122-5p	14.25	3.36 × 10^−5^
	slu-mir-200b-3p	5.11	3.25 × 10^−2^
	slu-mir-182-5p	4.79	3.29 × 10^−2^
*Enriched in fingerling 35 g*			
	slu-mir-1388-3p	6.68	2.45 × 10^−4^
	slu-mir-730-5p	5.32	5.11 × 10^−4^
	slu-mir-7132-5p	5.15	5.11 × 10^−4^
	slu-mir-731-5p	3.23	4.14 × 10^−2^
	slu-mir-125-1-3-5p	3.22	1.92 × 10^−2^
	slu-mir-99b-5p	3.13	2.50 × 10^−2^
Head kidney versus all other adult tissues			
*Enriched in head kidney*			
	slu-mir-10b-5p	165.58	1.77 × 10^−9^
	slu-mir-551-3p	55.04	3.44 × 10^−3^
	slu-mir-10c-5p	6.90	3.89 × 10^−2^
Spleen versus all other adult tissues			
*Enriched in spleen*			
	slu-mir-202-5p	24.59	2.03 × 10^−2^
	slu-mir-187-3p	20.70	3.52 × 10^−2^
	slu-mir-2187-3p	18.55	6.82 × 10^−4^
Gills versus all other adult tissues			
*Enriched in gills*			
	slu-mir-205-5p	585.85	3.67 × 10^−14^
	slu-mir-203b-3p	579.26	1.08 × 10^−18^
	slu-mir-725-3p	447.41	1.44 × 10^−14^
	slu-mir-203a-3p	398.99	2.42 × 10^−40^
	slu-mir-206-3p	144.42	2.29 × 10^−27^
	slu-mir-205-3p	115.31	1.45 × 10^−15^
	slu-mir-184ab-3p	73.51	3.44 × 10^−11^
	slu-mir-190b-5p	42.67	2.76 × 10^−6^
	slu-mir-375-3p	32.95	2.86 × 10^−2^
	slu-mir-182-5p	25.44	2.51 × 10^−17^

## Data Availability

All sequenced samples have been submitted to the NCBI Sequence Read Archive Centre (SRA) (https://www.ncbi.nlm.nih.gov/sra (accessed on 30 May 2023)) under Bioproject PRJNA977610.

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
