# Peer review of "The Discovery and Characterization of Conserved and Novel miRNAs in the Different Developmental Stages and Organs of Pikeperch (Sander lucioperca)"

_ijms, 2023, doi:10.3390/ijms25010189_

Round 1
Reviewer 1 Report
Comments and Suggestions for Authors
The micro RNA (miRNA) is novel non-coding RNA which exerts regulatory functions in cells. The research on the miRNA is helpful for life science. However, researches of the miRNA in aquaculture species are rare. This research can improve the knowledge of the miRNA in aquaculture species. After reading your manuscript, I found several problems. The following are my comments.
1. In introduction, please detail the regulatory functions of miRNAs on growth, immune and stress responses. Authors can exhibit results of those miRNA function researches in introduction. That is helpful for readers.
2. In line 66, please detail the bottlenecks in pikeperch. And it is better to provide information that the miRNA can break these bottlenecks.
3. The marker assisted selection was mentioned in line 67. The information of the miRNA using in marker assisted selection should be also provided in the introduction section.
4. The introduction of pikeperch is not enough. More characteristics of pikeperch should be provided in the introduction section, such as life habit, nutrient value, marketable value, and so on.
5. In figure 3, the font size of “Gills” is different from “Liver” and “Head Kidney”. Please adjust the font size to be consistent. And the “Gills” should be “Gill”, consisting to “Liver” and “Head Kidney”.
Comments on the Quality of English Language
no.
Reviewer 2 Report
Comments and Suggestions for Authors
Please see the attachment.

Comments on the Quality of English Language
No
Round 2
Reviewer 1 Report
Comments and Suggestions for Authors
It is ok for publication.